# Rapid Oxford Nanopore Technologies MinION Sequencing Workflow for *Campylobacter* *jejuni* Identification in Broilers on Site—A Proof-of-Concept Study

**DOI:** 10.3390/ani12162065

**Published:** 2022-08-13

**Authors:** Clara Marin, Francisco Marco-Jiménez, Llucia Martínez-Priego, Griselda De Marco-Romero, Vicente Soriano-Chirona, Laura Lorenzo-Rebenaque, Giuseppe D’Auria

**Affiliations:** 1Institute of Biomedical Sciences, Veterinary Faculty, Universidad Cardenal Herrera-CEU, 46113 Alfara del Patriarca, Spain; 2Institute of Science and Animal Technology, Universitat Politècnica de València, 46022 Valencia, Spain; 3Servicio de Secuenciación y Bioinformática, Fundación Para el Fomento de la Investigación Sanitaria y Biomédica de la Comunidad Valenciana (FISABIO-Salud Pública), 46022 València, Spain

**Keywords:** poultry, foodborne, 16S RNA, microbiota, Bento Lab

## Abstract

**Simple Summary:**

Current culture methods for *Campylobacter* enumeration in poultry meat require at least five days. To improve this outcome, we have assembled a portable sequencing kit consisting of the Bento Lab and the MinION and developed a workflow for on-farm use that can detect and report the presence of *Campylobacter* in less than five hours from sampling to data. In addition, this workflow allows for the determination of microbiota profiles within those five hours. Overall, this workflow and approach can be helpful in the control of zoonotic agents, with a particular focus on poultry meat, where it takes less than 24 h from the time the animals arrive at the slaughterhouse to the time the carcasses are available on the supermarket shelves.

**Abstract:**

*Campylobacter* is recognised as one of the most important foodborne bacteria, with a worldwide health and socioeconomic impact. This bacterium is one of the most important zoonotic players in poultry, where efficient and fast detection methods are required. Current official culture methods for *Campylobacter* enumeration in poultry usually include >44 h of culture and >72 h for identification, thus requiring at least five working shifts (ISO/TS 10272-2:2017). Here, we have assembled a portable sequencing kit composed of the Bento Lab and the MinION and developed a workflow for on-site farm use that is able to detect and report the presence of *Campylobacter* from caecal samples in less than five hours from sampling time, as well as the relationship of *Campylobacter* with other caecal microbes. Beyond that, our workflow may offer a cost-effective and practical method of microbiologically monitoring poultry at the farm. These results would demonstrate the possibility of carrying out rapid on-site screening to monitor the health status of the poultry farm/flock during the production chain.

## 1. Introduction

Since 2005, *Campylobacter* has been the most commonly diagnosed zoonotic pathogen in the European Union (EU), and one of the major causes of foodborne illness worldwide [1,2]. Poultry is considered the main reservoir of the bacteria, with the consumption of contaminated and/or inadequately prepared poultry products as the main source of outbreaks in humans [1,3,4]. Thus, in January 2018, the EU established Regulation (EC) No 1495/2017, to set a process hygiene criterion for *Campylobacter* control at slaughterhouses, and began to monitor the bacterial burden, with the limit of 1000 CFU/g of neck skin. Despite this control, approximately 220,000 cases of *Campylobacter* from meat consumption continue to occur every year [1]. The use of prevalence target *C. jejuni* models to assess the quantitative microbiological risk estimated that establishing targets of 25% and 5% between broiler flocks’ prevalence would lower the risk of human infection by chicken consumption to 50% and 90%, respectively [5,6]. Consequently, there is a need for more practical, suitable, and cost-effective techniques for routine diagnosis to determine the status of the flocks in a short turnaround time and with the highest sensitivity, respectively [7,8], to maintain the control of zoonotic agents that can contaminate food of animal origin, posing a risk to consumers’ health. In addition, it is important to choose the best moment to assess the *Campylobacter* status of the flock. There are two key moments in the broiler production cycle (14 and 42 days of age), because 14 d of age is when the broiler immune system starts to mature and the 42 d of age is when the animals leave for the slaughterhouse [9]. For this reason, any decision taken at these two crucial moments can directly influence the broiler production cycle and the final product.

In recent years, researchers have been implementing molecular techniques to develop new tools to assist scientists in epidemiological and ecological studies to control the common foodborne bacteria linked to poultry products associated with human gastroenteritis [10,11,12,13,14,15,16,17,18,19,20,21,22,23,24,25]. To this end, impressive progress has been made in next-generation sequencing (NGS), which has allowed for a more comprehensive characterisation of microbiota ecology in poultry [16,26,27,28,29]. Despite their availability, the types of equipment and analyses required for molecular techniques make the technology unavailable to the primary sector at present [30,31]. Related to this, a small and portable device called MinION (MinION, Oxford Nanopore Technologies, Oxford, UK) that applies nanopore sequencing technology to nucleic acid analyses, with far reaching applications including real-time bacterial metagenomic community analysis, is currently being used in important microbiological studies [32,33,34,35]. In the context of rapid identification of poultry-borne pathogens, this ability to yield information at the field level would be revolutionary for decision making. Thus, on-site sequencing applications may be an essential option for safeguarding the food supply [7,23,36]. To the best of our knowledge, NGS has not been applied on site for *Campylobacter* detection on the farm. Thus, employing this portable handheld real-time sequencing platform, the MinION, combined with a portable molecular biology lab, the Bento Lab [37], we conducted a proof-of-concept study to compare this rapid screening method based on 16S rDNA amplicon sequencing workflow with the official methodology in EU (ISO/TS 10272-2:2017) to detect *Campylobacter* and the relationship of *Campylobacter* with other caecal microbes at two sampling times at the farm level.

## 2. Materials and Methods

### 2.1. Experimental Design and Sampling

The experiment aimed to evaluate and validate a detection workflow under farm level conditions. The study was performed in the experimental poultry farm at the Centre for Research and Animal Technology (CITA-IVIA, in its Spanish acronym, Valencian Institute for Agrarian Research, Segorbe, Spain). A total of 42 one-day-old Ross broiler chicks were housed in six replicate pens (seven birds/pen) at 35 kg/m^2^ density. Briefly, the environmental temperature was gradually decreased from 32 °C (1 day) to 19 °C (42 days), in line with common practice in poultry production. All birds were allowed ad libitum access to feed and water. The nutritional programme consisted of two diets; starter (1 d to 21 d, Camperbroiler iniciación, Alimentación Animal Nanta, Valencia, Spain) and grower (21 d to 42 d, A-32 broiler, Alimentación Animal Nanta, Valencia, Spain). Birds and housing facilities were inspected daily for general health status, feed, water, temperature, mortality, and any unanticipated events.

### 2.2. Campylobacter Jejuni Infection

At day 11 days old, 20% of birds/pen were orally infected with *C. jejuni*. The strain was selected from a database of *Campylobacter* strains isolated in a longitudinal study of the whole poultry production cycle (breeders and their progeny) in the Valencia region (Eastern Spain) [38]. To generate the inoculum, an overnight inoculum was diluted to obtain a wavelength of 0.2 (OD 600, 10^8^ CFU/mL). Then, tenfold dilutions were made until a concentration of 10^6^ CFU/mL was achieved, and animals were infected with 100 μL (10^5^ CFU/bird). 

### 2.3. Sampling Collection

The *Campylobacter* status of the chicken houses was tested before the arrival of the animals taken from environmental samples with sterile swabs from the floor and walls from six different points of the house. All collected samples were analyzed within 24 h according to 10272-2:2017. 

To assess *Campylobacter* status of the animals upon arrival at the farm, 10 birds were randomly selected and euthanised. After necropsy, the pair of ceca were removed and placed in an individual sterile jar. Caeca samples were pooled into a composite sample for the detection of *Campylobacter*. On days 14 and 42, three birds per pen were randomly selected and euthanised. Individual caecum samples were obtained and placed in sterile pots. Briefly, the surface of each intact caecum was spray treated with 70% ethanol, as directed by Hansson et al. [39], before caeca contents were extracted. Then, the caeca content was removed and homogenized. To this end, the caeca content was placed in a sterile Petri dish and homogenized vigorously with a sterile swab until a homogenized product was obtained [40,41]. Then, pools of the three animals from the same pen were prepared and divided into two aliquots. One aliquot was analysed by the culture method and the other aliquot was analysed by 16S rDNA amplicon sequencing. Workflows for the ISO method and MinION nanopore sequencing performed are shown in Figure 1.

### 2.4. Culture Method

Enumeration of *Campylobacter* in samples was performed according to ISO/TS 10272-2:2017. As required by the ISO method, tenfold dilutions of samples on BPW were performed. Then, 0.1 mL from each inoculum was plated onto mCCDA (Modified Charcoal Cefoperazone Deoxycholate Agar, Oxoid, Dardilly, France) and incubated at 41.5 ± 1 °C in a micro-aerobic atmosphere (84% N_2_, 10% CO_2_, 6% O_2_) for 44 h ± 4 h. For further confirmation analysis, five *Campylobacter*-like colonies were purified on blood agar (AES La-boratories^®^, Bruz Cedex, France) at 41.5 ± 1 °C in a microaerobic atmosphere (84% N_2_, 10% CO_2_, 6% O_2_) over 24 h. Then, colony morphology and motility under dark field microscopy were evaluated. Confirmation of presumptive *Campylobacter* colonies was assessed by oxidase and catalase tests and plating at different temperatures and atmospheres (25 °C under aerobic conditions and 41.5 °C under microaerophilic conditions for 24 h) in Columbia blood agar (AES Laboratories). 

For *Campylobacter* speciation, isolated strains were plated on Columbia Blood Agar (Oxoid Ltd., Basingstoke, UK) and incubated at 41.5 °C for 44 ± 4 h in a modified atmosphere (5% O_2_, 85% N_2_, 10% CO_2_, CampyGen, Oxoid, Basingstoke, UK). Finally, the hippurate hydrolysis test (Oxoid, Madrid, Spain) was used to determine the species of the *Campylobacter* [38,42].

### 2.5. MinION Sequencing: Genomic DNA Extraction, Sequencing, and Analysis 

On-farm DNA analysis was performed using a mobile device called Bento Lab, which combines a portable PCR machine, a microcentrifuge, gel electrophoresis, and a transillumina-tor [37]. Genomic DNA was extracted from 1 g of caecal content with the Wizard Genomic DNA Purification Kit (Promega, Madison, WI, USA). After extraction, DNA was purified with one volume of Agencourt AMPure XP beads (Beckman Coulter, Inc, L’Hospitalet de Llobregat Barcelona, Spain). The quality and average size of genomic DNA were visualised by gel electrophoresis with a 1% agarose gel. For microbial community analysis, 10 ng of genomic DNA was used as an input for the Nanopore 16S Barcoding Kit (SQK-RAB204), which amplified the 16S rDNA gene by PCR using specific 16S primers 27F and 1492R containing barcodes and 5′ tags, following the manufacturer’s instructions. The sequencing was performed in a portable Min-ION sequencing device, with FLO-MIN106 flow cell (ONT Research, Oxford, UK). A total of 12 samples were sequenced in a run. Raw reads were base called into fast5 files with the Albacore program version 2.1.7 (ONT Research, Oxford, UK). The fast5 files were converted to fastq format using poretools version 0.6.0 [43]. For taxonomy analyses, taxonomy was assigned to each individual sequence read using the RDP classifier (v2.13, Trainset No: 18, 07/2020) [44]. The statistical analyses were performed in R software version 4.1.1 (R Core Team 2018) using the gdata [45], vegan [46], tydyr (https://CRAN.R-project.org/package=tidyr, accessed on 1 April 2022) packages. Reads’ time points were obtained from Nanopore fastq headers using the “start_time” field expressed as “YYYY-MM-DD HH:MM:SS”. This field value was extracted from every read and converted on a second-scale count. A taxonomy contingency table was built every 10 min using in-house R scripts.

ClustVis software was used to perform the PCA [47]. InteractiVenn software was used for Venn diagram construction [48]. Differences in relative abundances of taxonomic units between rearing days (14 vs. 42 days) were estimated using Bayesian interference. The sampling time was included as a treatment with two levels (14 vs. 42 days). The dendrogram obtained by Bayesian interference was created by 60,000 interactions of Markov chain Monte Carlo, with a burn-in period of 10,000 and saving only 1 of every 10 samples for inference. The parameters obtained from the marginal posterior distributions of the relative abundance between groups were the mean of the difference (D_14–24_; computed as 14–42), the probability of the difference being greater than 0 when D_14–24_ > 0 or lower than 0 when D_14–24_ < 0 (P_0_), and the highest posterior density region at 95% of probability (HPD95%). D_14–24_ estimated the mean of the differences between 14 and 42 traits, P_0_ estimated the probability of D_14–24_  ≠  0, and HPD95% estimated the accuracy. Statistical differences were considered if P_0_  >  0.8 (80%). Statistical analyses were computed with the Rabbit program developed by the Institute for Animal Science and Technology (Valencia, Spain). To determine the shortest MinION sequencing time that could generate sufficient data for *Campylobacter* taxa prediction, raw reads generated after 360 min of sequencing were divided into 10 min batches using read timestamps included in the read’s names during the data production per se. The generated and analyzed datasets are available at NCBI’s BioProject PRJNA814618.

### 2.6. Sequencing Costs

We also calculated the sequencing costs associated with MinION to examine how they compared with the conventional culture. Our calculations did not consider the entry cost of sequencing hardware (i.e., the price of a MinION sequencer, EUR 900). For MinION sequencing, we assumed that reagents and flow cells were purchased separately because this provides a more meaningful gauge, as the MinION starter pack is usually a one-time purchase for most users. In addition, the personal labour cost was excluded.

## 3. Results

During this study, a total of 12 caecal pools (6 per sampling time) were collected and processed through microbiological culture and sequencing. No clinical signs were ob-served during rearing, and the productive parameters obtained were in accordance with the breed standards. Environmental samples and day-old chick samples were confirmed as *Campylobacter* negative. During rearing, all caeca samples were positive for *Campylo*-bacter with both methods (Table 1). For the ISO 1027-2:2017 method, starting at the enumeration to identification, a total of four days was necessary to obtain the result. All samples were positive, regardless of the sampling day, with the mean number of *Campylobacter* being 7.4 log CFU/g on day 14 and 6.7 log CFU/g on day 42 (Table 1).

We found significant differences between 14 and 42 days of rearing in the number of *Campylobacter* (Table 2). The complete MinION sequencing process, starting from the DNA extraction to the extraction to a minimum limit of 239,988 reads, lasted 10.5 h (average of 20,000 reads by sample, Appendix A). Read length ranged from 1542.00 ± 241.50  nt to 1563.02 ± 243.71  nt, with an average of 1553.55 ± 237.55  nt (mean ± SD). As in the ISO 1027-2:2017 method, the mean relative abundance for *Campylobacter* was conditional by sampling day (Table 2), with the relative abundance for *Campylobacter* being less on day 42.

### 3.1. Taxonomic Diversity

Alpha diversity was assessed through species evenness and species richness measures. Rarefaction analysis indicated that the number of amplicon sequence variants (ASVs) per sample reached a maximum, indicating that the sequencing effort captured the diversity of the majority of the samples. The observed sequences and Chao1 alpha diversity indexes revealed a significant difference between the caecal microbiota depending on the age of the animals (155.2 ± 5.6 vs. 213.83 ± 5.6 for observed sequences, *p* ≤ 0.01 and 193.2 ± 13.3 vs. 284.5 ± 13.3 for Chao1, *p* ≤ 0.01, for 14- and 42-day-old chicks, respectively). Finally, to assess differences in microbiota between rearing sample times, beta diversity was analysed. To this end, principal coordinate analysis (PCoA) was conducted based on Bray–Curtis dissimilarity, and the results are shown in Figure 2. In both 14- and 42-day-old chicks, the six replicates from each rearing day clustered together. These results showed differences in the microbial community composition among the two broiler-rearing times.

### 3.2. Microbial Community Dynamics

The complete list of all bacterial taxa (average relative abundance of more than 2%) identified for each age group in the caecal content samples is provided in Table 3. Microbial taxa consistently present over time (core microbiome) represented by 22 genera (mainly belonging to the order *Clostridiales*, phylum *Firmicutes*) were identified in both sampling times with variations in the relative abundance (Table 3). The highest count of *Campylobacter* was found in 14-day-old broilers (1.36%, *p* = 1.00, HPD 95% [0.79, 1.94]).

### 3.3. Effect of Sequencing Time on *Campylobacter* spp. Identification

To investigate the potential robustness to detect *Campylobacter* spp. throughout the sequence time, we examined changes in the relative abundance of *Campylobacter* spp. every 10 min from 0 to 360 (Figure 3). There was a rapid increase in the relative abundance of *Campylobacter* spp. detection from 0 to 20 sequencing minutes, after which it was constant until 280 min, regardless of the rearing moment, with a relative abundance average of 1.78 ± 0.105% and 0.13 ± 0.037% for 14- and 42-day-old chick samples (mean ± standard deviation), respectively (Figure 3). After 320 min of sequencing, 14-day-old chick samples slightly increased the relative abundance of *Campylobacter*, while that of the 42-day-old chick samples slightly decreased. 

### 3.4. Sequencing Costs

Overall, the per sample costs for MinION sequencing were found to be approximately eighteen-fold more expensive than the conventional culture method (Table 4), costing an estimated EUR 75 per sample considering 12 samples per run. This cost can be further reduced down to about EUR 41 per sample pooling up to 24 barcodes.

## 4. Discussion

When broilers were artificially infected with *Campylobacter jejuni*, Firmicutes and Campilobacterota were the most dominant phyla in the caecal microbiota. Notably, the current study reports fast *Campylobacter* (and core microbiota) screening under farm conditions. The decision–action gap is crucial in intensive farming systems such as poultry production, where the high throughput of animal husbandry presents a high risk of developing and transmitting zoonotic agents [49,50,51]. *Campylobacter* infection is one of the the last century’s most widespread foodborne infectious diseases, and poultry has been considered the origin of 80% of human cases [52,53]. Thus, our study yields two main results. First, we can detect *Campylobacter* spp. in less than five hours, from sample collection to result, which has some important implications. Second, the workflow developed here shows remarkable potential for simultaneous detection of foodborne pathogens and provides an overview of the core microbiota that could be related at a given time to the health and welfare status of the animals. As on-farm detection represents the first step necessary to limit the introduction of *Campylobacter* into the food chain [54,55], our workflow can provide a cost-effective and practical method for on-farm real-time monitoring of microbial contamination of poultry. Although the costs of MinION sequencing seem high, they are generally inversely related to sample size [56]. One of the advantages of nanopore sequencing is that it is possible to stop sequencing as soon as sufficient data have been obtained and washed with a nuclease kit (100 euros for six washes), and the flow cell has been resued with other samples.

Based on the European Regulation, official *Campylobacter* detection in poultry must be carried out taking samples from neck skin during processing at the slaughterhouse and analysing them according to ISO 10272:2018 [53,55,56,57]. The process involves a ten-fold dilution of collected samples incubated at 41.5 °C in a microaerobic atmosphere for 48 h. Further characterisation was performed by morphology and motility and biochemical confirmation at different temperatures and atmospheres [57,58]. This technique makes these assays laborious, and they require several days of work (taking up to five days) [53,58,59]. The main problem is that the official samples are taken at the slaughterhouse, and the positive results have to be obtained as soon as possible to be able to make decisions about the destination of the flock [10,55]. It is important to note that, within just 24 h after the slaughter of the animals, the chickens are already on the supermarket shelves, ready to be consumed. Undoubtedly, it seems essential to assess the status of the flocks before they reach the slaughterhouse [30,60,61,62,63,64]. This is why molecular methods for same-day diagnosis of pathogenic bacteria are increasingly being introduced into daily laboratory routines [53,65,66]. Overall, 100% of samples were positive for both methods, regardless of animal age, requiring a total of five days from enumeration to confirmation for ISO 1027-2:2017, while the sequencing method required only five hours between extraction and sequencing. It is notable that, regardless of the detection method, the colonisation of *Campylobacter* depended on the age of the bird after the infection challenge, with a relevant reduction observed in 42-day samples, in line with the literature [67,68,69]. 

Nanopore MinION has been proposed as a fast, efficient, sensitive, and cost-effective sequencing method capable of enabling the identification of pathogens based on the core microbiome and differentially abundant taxa at the field level [35,70]. In this study, the real-time availability of the data generated by MinION showed that, after 20 min of sequencing time, the relative abundance of *Campylobacter* remained constant up to 320 min, with a similar pattern observed at both rearing ages. In addition, this workflow makes it possible to analyse several samples at the same time in order to obtain a general overview of the microbiota, which could be useful for routine surveillance and early detection of foodborne pathogens [36]. Beyond that, as the microbiota influences various physiological and behavioural processes, a healthy microbiota would be correlated with an animal’s fitness, making it possible to identify health and welfare deviations in the animals [71]. Though the workflow proposed here was based on data analysed after complete data collection, taxonomy sequences were obtained a posteriori, after the sequencing process finished, using per-sequence timestamps added by the nanopore sequencing system itself to each read. At the time of writing, the whole workflow can be easily transferred to an automated offline bioinformatics pipeline, as well as using the EPI2ME WIMP workflow: quantitative, real-time species identification from metagenomic samples. Taking our results together, the combination of mobile DNA barcoding with Bento Lab and MinION could be suitable for on-farm routine analysis of *Campylobacter* and other zoonotic pathogens, as the results can be obtained on the same day at the same site at which the samples have been taken [35,37].

## 5. Conclusions

Here, we proposed a MinION-based workflow for fast *Campylobacter* and other zoonotic pathogens’ screening using a Bento Lab–MinION combination for portable sequencing at the farm level. We have demonstrated that MinION-based workflow at the farm level is viable and highly accurate by comparing it with ISO/TS 10272-2:2017. We conclude that this method could provide a basis for future studies to help farmers implement active zoonotic bacterial screening in real-time diagnostics to take measures that will reduce the prevalence, which is indispensable for active surveillance. 

## Figures and Tables

**Figure 1 animals-12-02065-f001:**
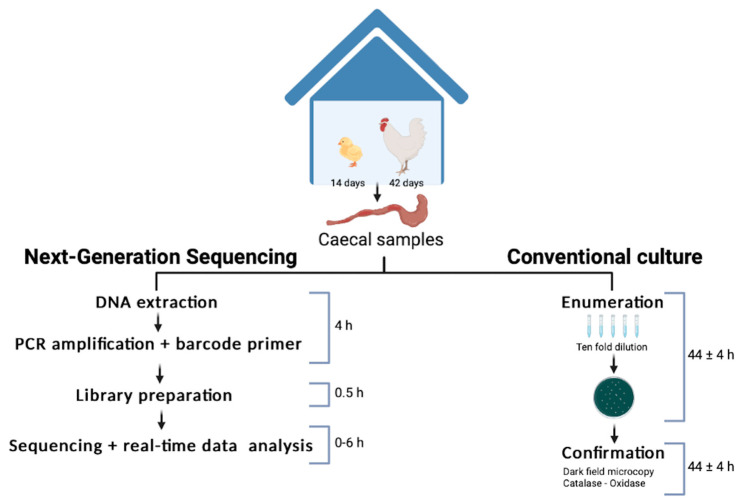
Timeline for the rapid method developed at farm level compared with the current culture detection method. The timeline is based on the work of one laboratory technician.

**Figure 2 animals-12-02065-f002:**
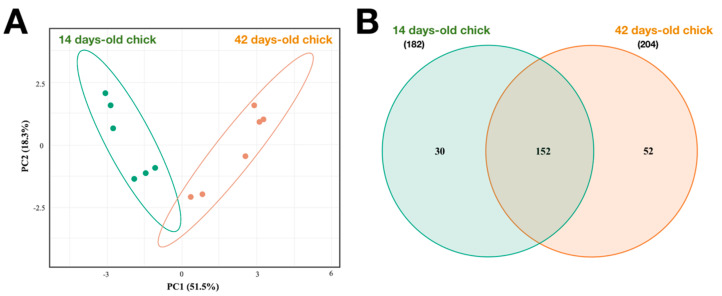
Principal component analysis (PCoA) and Venn diagram summarising differential microbiota community in caecal samples across 14 and 42 days of rearing cycle. (**A**) Evaluation of the beta diversity based on Bray–Curtis dissimilarity between growing stages (14- and 42-day-old chicks) represented by a PCoA graphic. (**B**) Venn diagrams showing the distribution of shared and unique sequences assigned at the genus level among the different growing stages (14- and 42-day-old chicks).

**Figure 3 animals-12-02065-f003:**
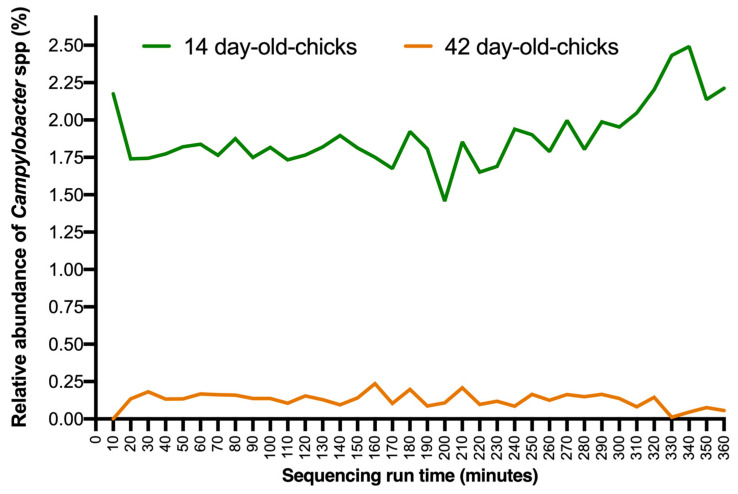
Relative abundance of *Campylobacter* sequences in caecal samples generated by 10 min over six hours of sequence time using the MinION sequencing device.

**Table 1 animals-12-02065-t001:** Descriptive analysis of the *Campylobacter* load in caecal samples by the conventional cultured method according ISO 10272:2018 and next-generation sequencing method by MinION nanopore technology. Caecal samples were compared at 14 and 42 days of rearing cycle.

Day of Rearing Cycle	Pool	Pen	Method	
			Conventional Culture	Sequencing
			Enumeration (Log10 CFU/g)	Relative Abundance * (%)
14	1	1	8.4	1.73
	2	2	8.0	1.86
	3	3	6.2	1.66
	4	4	8.2	3.24
	5	5	8.0	1.82
	6	6	5.6	0.64
Mean ± SD			7.4 ± 0.48	1.8 ± 0.33
42	7	1	6.7	0.06
	8	2	7.2	0.07
	9	3	5.9	0.04
	10	4	6.6	0.30
	11	5	7.0	0.07
	12	6	7.1	0.14
Mean ± SD			6.7 ± 0.19	0.1 ± 0.04

* Data are expressed as mean ± standard deviation (mean ± SD).

**Table 2 animals-12-02065-t002:** Bayesian analyses of the *Campylobacter* load in caecal samples by the conventional cultured method according ISO 10272:2018 and next-generation sequencing method by MinION nanopore technology. Caecal samples were compared at 14 and 42 days of the rearing cycle.

Method	D_14–42_	P_0_	HPD95%
Conventional culture (Log10 CFU/g)	0.65	0.86	−0.60, 1.95
Next-generation sequencing (%)	1.72	1.00	0.83, 2.55

D_14–42_ = mean of the difference between 14 and 42 days of rearing (median of the marginal posterior distribution of the difference between the 14 and 42 days of rearing). P_0_ = probability of the difference (D_14–42_) being greater than 0 when D_14–42_ > 0 or lower than 0 when D_14–42_ < 0. HPD95% = the highest posterior density region at 95% of probability. Statistical differences were assumed if |D_14–42_| surpass the R value and its P_0_ > 0.80.

**Table 3 animals-12-02065-t003:** Taxonomic profiles at genus level and Bayesian analyses of the relative abundance comprising more than 2% of the total bacterial sequences in at least one of the different ages of rearing (14- and 42-day-old chicks).

Phylum	Family	Genus	Day 14 (%)	Day 42 (%)	D_14–42_ (%)	P_0_	HPD 95%
*Campilobacterota*	*Campylobacteraceae*	*Campylobacter*	1.44	0.08	1.36	1.00	0.79, 1.94
*Firmicutes*	*Lactobacillaceae*	*Holzapfelia*	0.69	3.06	−2.36	0.99	−4.04, −0.64
		*Limosilactobacillus*	0.48	2.86	−2.37	0.97	−4.75, 0.07
		*Paralactobacillus*	2.24	8.62	−6.35	0.98	−12.5, −0.64
	*Eubacteriaceae*	*Intestinibacillus*	2.20	0.54	1.68	0.97	−0.06, 3.42
	*Lachnospiraceae*	*Bariatricus*	12.29	6.83	5.52	0.97	0.03, 11.57
		*Cuneatibacter*	0.56	2.39	−1.83	0.97	−3.72, 0.11
		*Faecalimonas*	4.90	3.53	1.38	0.95	−0.27, 3.08
		*Fusicatenibacter*	3.08	2.85	0.24	0.64	−1.29, 1.68
		*Lachnotalea*	2.22	1.23	0.99	0.99	0.17, 1.82
		*Robinsoniella*	3.20	2.85	0.36	0.70	−1.07, 1.95
		*Sellimonas*	2.30	1.08	1.23	1.00	0.71, 1.76
	*Ruminococcaceae*	*Agathobaculum*	3.58	1.15	2.40	0.96	−0.14, 5.29
		*Dysosmobacter*	1.86	2.04	−0.17	0.60	−1.74, 1.43
		*Faecalibacterium*	4.12	8.04	−3.87	0.91	−10.03, 1.86
		*Lawsonibacter*	3.60	2.55	1.06	0.98	0.06, 2.06
		*Negativibacillus*	2.89	0.63	2.26	1.00	0.84, 3.81
		*Neglecta*	3.32	2.02	1.30	0.94	−0.39, 3.05
		*Petroclostridium*	1.01	2.90	−1.90	1.00	−3.04, −0.61
		*Pseudoclostridium*	2.17	2.81	−0.64	0.71	−3.29, 1.82
		*Ruthenibacterium*	6.17	0.39	1.79	0.92	−0.90, 4.51
		*Subdoligranulum*	1.13	2.36	−1.23	0.99	−2.09, −0.33

D_14–42_ = mean of the difference between 14 and 42 days of rearing (median of the marginal posterior distribution of the difference between the 14 and 42 days of rearing). P_0_ = probability of the difference (D_14–42_) being greater than 0 when D_14–42_ > 0 or lower than 0 when D_14–42_ < 0. HPD95% = the highest posterior density region at 95% of probability. Statistical differences were assumed if |D_14–42_| surpass the R value and its P_0_ > 0.80.

**Table 4 animals-12-02065-t004:** Cost comparison between MinION sequencing and conventional culture for this study.

Method		Package Cost (EUR)	Per Sample Cost (EUR)
**MinION ***			
DNA extraction and PCR amplification	Wizard Genomic DNA Purification Kit (Promega, 100 samples)	203	2.03
Native barcoding expansion	16S Barcoding Kit 1-24 (for up to 144 samples)	813	5.65
Sequencing	FLO-MIN106 flow cell (1 flow)	810	67.50
		Total	75.18
**Conventional culture**			
Enumeration	Buffered Peptone Water (BPW, 500 g)	50	0.22
	Modified Charcoal Cefoperazone Deoxycholate Agar (mCCDA, Oxoid, Dardilly, France)	1	1.0
Confirmation	Blood agar plate (AES La-boratories^®^, Bruz Cedex, France)	1	3.0
		Total	4.22

* Publicly available pricing was used for MinION sequencing costs (https://store.nanoporetech.com/eu/16s-barcoding-kit-1-24.html, accessed on 1 June 2022).

## Data Availability

Bioproject: PRJNA814618, BioSample: SAMN26561571.

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
