# Peer review of "Rapid Oxford Nanopore Technologies MinION Sequencing Workflow for Campylobacter jejuni Identification in Broilers on Site—A Proof-of-Concept Study"

_animals, 2022, doi:10.3390/ani12162065_

Round 1
Reviewer 1 Report
The authors present a rapid method for screening for Campylobacter using a portable laboratory with a metabarcoding sequencing approach. Overall, the method has merit for both detection of Campylobacter spp, and potentially other foodborne pathogens, as well as for evaluation chicken microbiome status. I think that a major gap in this work is that there were no negative controls presented. Is there potential for false positive detection of Campylobacters? Given the high similarity of 16S sequences among closely related genera, and high error rate of the ONT sequencing technologies, I would worry about potential false positives. The authors show differences in microbiome profiles at two time points. Would this be the same in uninfected birds? I also wonder why caecal samples are used rather than feces. Would feces not be a better target if the intention is to deploy this system at the farm level? Authors should include discussion of potential cost of the approach. What would be the advantages relative to cheaper technologies such PCR? I understand that it is nice to have microbiome information, but are there actions associated with perturbations in these profiles?
Seemed that some details were missing. For example, approximately how many cells of campylobacters were given to the birds? It was not clear where the environmental samples were taken.
I don’t think there are enough details about sequence analysis provided to reproduce the work. Perhaps precise commands for each of the software tools could be provided as a supplemental table.
Generally:
The manuscript could use further extensive editorial reviews. For example:
Use “sampling time” rather than “sampling moment”
Spell out numbers less than 10
Remove unnecessary hyphens within words (e.g. Line 58 “allowed”)
Use “MinION” throughout the text (sometimes MinIon is used, sometimes hyphens)
Specific comments:
Lines 71-72: consider rewording- “…based on a 16S rDNA amplicon sequencing workflow to detect Campylobacter at the farm level”
Line 84: …at the farm level…
Line 88: remove and from “under and farm environmental…”
Line 103: Not clear what dose was given to the birds? 1 mL containing 10e5 cells?
Line 110: “tested by a culture method”
Line 120: “analysed by the culture method” , remove hyphen from analysed
Line 125: For figure 1, it would be good to have a more descriptive/succinct name for the rapid method—maybe “rapid metabarcoding method”?
Line 132 “in BPW”/line 133-“and 0.1mL from…”
Line 135:-N2/CO2/O2 –2 should be in subscript (check for this in rest of document)
Line 139: change “suspicious” to “presumptive Campylobacter” colonies
Line 157: My understanding is that Albacore converts the fast5 files to fastq. The fast5 files are the raw output of the MinION? More details about the exact commands used/tools within packages used would be useful to enable reproduction of the work done here.
Line 164: period missing at end of sentence.
Line 196: What is meant by “Next sequencing method”
Table 2: I do not see an R variable anywhere, not sure if this was meant to be included
Line 252: Abundance
Line 257: Table 3 -> comprising? Not compromising; also does not have R variable included but refers to it in notes below?
Line 284: at the farm level
Lines 308-309: “It is important to note that in just 48 hours after the slaughter of the animals, the chickens are already on the supermarket shelves, ready to be consumed.”
o Inconsistent with the summary on first page:
o Lines 21-22: “Overall, this workflow and approach can be helpful in the control 20 of zoonotic agents, with a particular focus on poultry meat, where it takes less than 24 hours from the time the animals arrive at the slaughterhouse to the time the carcasses are available on the supermarket shelves.”
Author Response
Dear Reviewer
The authors of this study thank you very much for your comments and suggestions for manuscript improvement. All of them have been detailed below, and the authors of the manuscript hope the answers to your questions and the improvement of the document following your indications will be to your agreement.
I think that a major gap in this work is that there were no negative controls presented. Is there potential for false positive detection of Campylobacters?
The authors of this manuscript fully understand the reviewer's comment; however, fine-tuning the MinIon was not the aim of the study, as it is a methodology that is routinely used in human health (a technique that has already been fine-tuned in numerous hospitals with the same samples but of human origin). The aim of this study has always been to work with real field samples from animals infected with a known concentration of Campylobacter to compare the official methodology used in the EU (ISO) and the MinIon (Proof of concept). Therefore, the "control" of our study was the analyses carried out through the ISO standard, which is the one taken into account to determine whether or not the products derived from these animals are suitable for human consumption. In addition, the DNA extractions, the ISO standard technique and the MinIon technique always had the internal controls that all laboratory techniques must-have, both positive and negative. Whenever Campylobacter was detected with the MinIon, it was also detected with the ISO standard. Following the reviewer's comment, we have clarified this point in the manuscript to avoid misunderstandings.
The title has been modified for better understanding “Rapid Oxford Nanopore Technologies MinION sequencing workflow for Campylobacter spp. identification in broilers on-site - a proof of concept study” (Line 4).
The sentence “Current official culture methods for Campylobacter enumeration in poultry usually include >44 h of culture and >72 h for identification, thus requiring at least five working shifts (ISO/TS 10272-2:2017)” has been included in the abstract for better understanding (Lines 26-28).
The objective has been re-written: So, employing this portable handheld real-time sequencing platform, the MinION, combined with a portable molecular biology lab, the Bento Lab [36], we conducted a proof of concept to compare this rapid screening method based on 16S rDNA amplicon sequencing workflow with the official methodology in EU (ISO/TS 10272-2:2017) to detect Campylobacter at two sampling times at farm level (Lines 78-82)
Given the high similarity of 16S sequences among closely related genera, and high error rate of the ONT sequencing technologies, I would worry about potential false positives. The authors show differences in microbiome profiles at two time points. Would this be the same in uninfected birds?
Effectively, many articles have been published describing the normal microbiota of a bird. However, going beyond the description or trying to interpret any variation is not easy. Therefore, the aim of this work has always been proof of concept, i.e. to apply a technique that has already been developed and applied to human intestinal samples. As indicated in the previous section, negative and positive controls were included in both diagnostic techniques to avoid false positives or negatives, as indicated by the reviewer. In the same way, a direct correlation between the two techniques compared was observed.
I also wonder why caecal samples are used rather than feces. Would feces not be a better target if the intention is to deploy this system at the farm level?
The faeces collection is efficient for Salmonella control (in fact, it is the official sample, together with dust, of the National Salmonella Control Plans in Europe), as it is a bacterium that remains viable for a very long time in the environment. In the case of Campylobacter, the epidemiology is very different, it is transmitted horizontally, but its survival in the environment is very short. Therefore, bird-to-bird contact is essential for transmission. When the veterinarian/visitor goes to the poultry farm, it is a common practice to select some healthy birds and/or birds with some symptoms to do routine necropsies and ensure that the health status of the flock of birds is correct. At this time, when the necropsy is performed, the optimal sample can be obtained from the caecum. This is the organ where the microbiota is richest in poultry and where sampling is not an extra expense.
Authors should include discussion of potential cost of the approach. What would be the advantages relative to cheaper technologies such PCR?
The fact that it is cheaper, considering that the cost would be assumed by the farmer/poultry company, is significant for the sector, which has also shown great interest in the subject. The fact that results are obtained more quickly and can be included in the routine work of a farm (necropsies) makes this technique very interesting and sustainable over time, as happens in poultry farming with routine serology to assess other microorganisms or the efficacy of a vaccine.
The information related to the cost of both techniques has been included in a specific section within the material and methods (Lines 207-214) and results (311-316). In addition, a comparative table has also been included (Table 4). (Lines 317-320)
I understand that it is nice to have microbiome information, but are there actions associated with perturbations in these profiles?
Each farm has unique characteristics, depending on management, location, feeding etc. With the MinIon technology, the microbiome's evolution could be assessed during rearing, and it would be possible to see how they evolve. By studying these evolutions, it could even be seen that some health problem occurs later when a microbial population increases or decreases. This would help to anticipate the problem and thus be able to make quick decisions in situ, e.g. application of acids in water, turning over the litter, increasing ventilation, etc.
Seemed that some details were missing. For example, approximately how many cells of Campylobacters were given to the birds?
According with the reviewer's suggestions, the material and methods section Campylobacter jejuni infection, has been re-written (Lines 118-113):
“At day 11 days old, 20% of birds/pen were orally infected with C. jejuni. The strain was selected from a database of Campylobacter strains isolated in a longitudinal study of the whole poultry production cycle (breeders and their progeny) in the Valencia region (Eastern Spain) [37]. To generate the inoculum, an overnight inoculum was diluted to obtain a wavelength of 0.2 (OD 600, 108 CFU/mL). Then, ten-fold dilutions were made until a concentration of 106 CFU/mL, and animals were infected with 100 μL (105 CFU/bird)”.
It was not clear where the environmental samples were taken.
According with the reviewer's suggestions, the material and methods section sampling collection, has been re-written (Lines 118-121):
Campylobacter status of the chicken houses was tested before the arrival of the animals taken environmental samples with sterile swabs from the floor and walls from 6 different points of the house. All collected samples were analyzed within 24h according to 10272-2:2017.
I don’t think there are enough details about sequence analysis provided to reproduce the work. Perhaps precise commands for each of the software tools could be provided as a supplemental table.
According with the reviewer's suggestions, the MinION sequencing: Genomic DNA extraction, sequencing and analysis section, has been completed (Lines 181-185) as:….tydyr (https://CRAN.R-project.org/package=tidyr) packages. Reads time points were obtained from Nanopore fastq headers using the “start_time” field expressed as “YYYY-MM-DD HH:MM:SS”. This field value was extracted from every read and converted on a seconds scale counts. A taxonomy contingency table was built every 10 minutes time span using in-house R scripts.
Generally:
The manuscript could use further extensive editorial reviews. For example:
Point 1: Use “sampling time” rather than “sampling moment”
Response 1: According with the reviewer suggestion, the term has been changed.
Point 2: Spell out numbers less than 10
Response 2: Numbers less than 10 has been spelled out
Point 3: Remove unnecessary hyphens within words (e.g. Line 58 “allowed”)
Response 3: Unnecessary hyphens have been corrected
Point 4: Use “MinION” throughout the text (sometimes MinIon is used, sometimes hyphens)
Response 4: The term has been corrected
Specific comments:
Point 5: Lines 71-72: consider rewording- “…based on a 16S rDNA amplicon sequencing workflow to detect Campylobacter at the farm level”
Point 6: Line 84: …at the farm level…
Response 6: The sentence has been rewritten (Lines 93: under farm level conditions)
Point 7: Line 88: remove and from “under and farm environmental…”
Response 7: The sentence has been removed.
Point 8: Line 103: Not clear what dose was given to the birds? 1 mL containing 10e5 cells?
Response 8: The experimental infection was done with 105 CFU/bird.
Point 9: Line 110: “tested by a culture method”
Response 9: The sentence has been rewritten (were tested in accordance with ISO 10272-2:2017).
Point 10: Line 120: “analysed by the culture method” , remove hyphen from analysed
Response 10: Hyphen has been removed
Point 11: Line 125: For figure 1, it would be good to have a more descriptive/succinct name for the rapid method—maybe “rapid metabarcoding method”?
Point 12: Line 132 “in BPW”/line 133-“and 0.1mL from…”
Response 12: The sentence has been rewritten (Lines 145-146: As required by the ISO method, ten-fold dilutions of samples on BPW were performed. Then, 0.1 mL from each inoculum was plated onto mCCDA).
Point 13: Line 135: -N2/CO2/O2 –2 should be in subscript (check for this in rest of document)
Response 13: Superscripts have been corrected.
Point 14: Line 139: change “suspicious” to “presumptive Campylobacter” colonies
Response 14: The term has been changed (Line 152).
Point 15: Line 157: My understanding is that Albacore converts the fast5 files to fastq. The fast5 files are the raw output of the MinION? More details about the exact commands used/tools within packages used would be useful to enable reproduction of the work done here.
Response 15: After sequencing on the MinION the corresponding fastA files were exported. Ribosomal Database Project (RDP) classifier version 2.13 was used for taxonomic assignment of the fastA files. All data analysis was carried out with R (v. 4.1.1). Data manipulation was performed with “gdata” package. Rarefraction and bacterial community diversities were calculated with the ‘vegan’ package.
Point 16: Line 164: period missing at end of sentence.
Response 16: Period has been added (Line 192).
Point 17: Line 196: What is meant by “Next sequencing method”
Response 17: The term has been corrected.
Point 18: Table 2: I do not see an R variable anywhere, not sure if this was meant to be included
Response 18: The relevant value (R) has not been included in the table. It does not provide more information and is thus more accessible for people not used to Bayesian analysis). It has been removed from the bottom of the table.
Point 19: Line 252: Abundance
Response 19: The term has been corrected (Line 283: counts).
Point 20: Line 257: Table 3 -> comprising? Not compromising; also does not have R variable included but refers to it in notes below?
Point 20: It is a mistake. The word comprising has been included. The relevant value has not been included in the table (it does not provide more information and is thus more accessible for people not used to Bayesian analysis). It has been removed from the bottom of the table.
Point 21: Line 284: at the farm level
Response 21: The sentence has been rewritten (Lines 322-324): When broilers were artificially infected with Campylobacter jejuni, Firmicutes and Campilobacterota are the most dominant phyla in the caecal microbiota. Notably, the current study reports fast Campylobacter (and core microbiota) screening under farm conditions.
Point 22: Lines 308-309: “It is important to note that in just 48 hours after the slaughter of the animals, the chickens are already on the supermarket shelves, ready to be consumed.”
- Inconsistent with the summary on first page:
- Lines 21-22: “Overall, this workflow and approach can be helpful in the control 20 of zoonotic agents, with a particular focus on poultry meat, where it takes less than 24 hours from the time the animals arrive at the slaughterhouse to the time the carcasses are available on the supermarket shelves.”
Response 22: According to the reviewer comment, the mistake has been solved.
Reviewer 2 Report
GENERAL COMMENTS
The manuscript addresses the important issue of rapid detection of Campylobacter species on farm. In this regard, the authors report on the use of a Rapid Oxford Nanopore Technologies MinION sequencing workflow.
Following are general comments on the manuscript:
1. Control treatment birds should have been included in the experimental design (i.e. birds not inoculated with C. jejuni).
2. Examination of Campylobacter at a species level of resolution would have been beneficial, and is important for potential adoption of the technology. Although it was not specified, presumably the near complete 16S rRNA gene was sequenced, and thus allows for species level determination.
3. The information provided on the diversity and structure of caecal communities in birds at 14 and 42 days-of-age is of marginal value, and outside the scope of the study.
4. The number of samples examined is small (n=6) and limited to birds inoculated with C. jejuni. Expanding the study to actual production samples would be beneficial.
5. Information on the relative cost (and merits) of the technology relative to other strategies (e.g. quantitative PCR) would be useful to include.
SPECIFIC COMMENTS
L37. Specify the Campylobacter species (presumably C. jejuni) here and throughout.
L58. Anomalous hyphen. This was noted elsewhere as well.
L76. Were requisite biosafety and biosecurity standards met? This should be specified.
L88. Superscript m2.
L100. Indicating “spp.” in the subheading is misleading. I was expecting to see information on speciation of Campylobacter.
L102. It would be beneficial to include additional information on the strain of C. jejuni used. Moreover, determination that the strain recovered was the same as the inoculation strain should be determined (i.e. fingerprinted).
L103. Additional information on how the inoculum was generated should be included.
L117. Spraying the caeca with ethanol may kill C. jejuni, but it will not kill endospores. Thus, the sentence should be reworded.
L118. How the samples were homogenized should be specified. Were beads used? This is crucial to guard against bias due to differential homogenization of gram negative vs gram positive taxa.
L136. Italicize Campylobacter here and throughout.
L139. Going beyond oxidase and catalase tests is necessary. For example, utilization of taxon-specific PCR.
L179. “serotype prediction”?
L269. Similarly to L100, this subheading is misleading.
L343. “This workflow could help farmers …”. May be so, but the study scope could have better addressed this (see general comment above).
Author Response
Dear Reviewer
Thank you very much for all your comments and suggestions for improvement. All of them have been detailed below, and the authors of the manuscript hope that both the answers to your questions and the improvement of the document following your indications will be to your agreement.
Following are general comments on the manuscript:
Point 1: Control treatment birds should have been included in the experimental design (i.e. birds not inoculated with C. jejuni).
Response 1: The authors of this manuscript fully understand the reviewer's comment; however, fine-tuning the MinIon was not the aim of the study, as it is a methodology that is routinely used in human health (a technique that has already been fine-tuned in numerous hospitals with the same samples but of human origin). This study has always worked with real field samples from animals infected with a known concentration of Campylobacter to compare the official methodology used in the EU (ISO) and the MinIon (Proof of concept). Therefore, the "control" of our study was the analyses carried out through the ISO standard, which is the one taken into account to determine whether or not the products derived from these animals are suitable for human consumption. In addition, the DNA extractions, the ISO standard technique and the MinIon technique always had the internal controls that all laboratory techniques must-have, both positive and negative. Whenever Campylobacter was detected with the MinIon, it was also detected with the ISO standard. Following the reviewer's comment, we have clarified this point in the manuscript to avoid misunderstandings.The title has been modified for better understanding “Rapid Oxford Nanopore Technologies MinION sequencing workflow for Campylobacter jejuni identification in broilers on-site - a proof of concept study” (Line 3).
The sentence “Current official culture methods for Campylobacter enumeration in poultry usually include >44 h of culture and >72 h for identification, thus requiring at least five working shifts (ISO/TS 10272-2:2017)” has been included in the abstract for better understanding (Line 28).
The objective has been re-written: So, employing this portable handheld real-time sequencing platform, the MinION, combined with a portable molecular biology lab, the Bento Lab [36], we conducted a proof of concept to compare this rapid screening method based on 16S rDNA amplicon sequencing workflow with the official methodology in EU (ISO/TS 10272-2:2017) to detect Campylobacter at two sampling times at farm level (Lines 78-82)
Point 2: Examination of Campylobacter at a species level of resolution would have been beneficial, and is important for potential adoption of the technology. Although it was not specified, presumably the near complete 16S rRNA gene was sequenced, and thus allows for species level determination.
Response 2: The authors of this study fully agree with the reviewer's comment. MinION™platform permits a taxonomic identification at the species level, indeed, its reads should provide better specificity than other sequencing methods such as Illumina; however, the ability to discriminate between closely related species is still lower. Indeed, in other previously published studies, we have used different molecular techniques for Campylobacter identification (Marin et al., 2020. Doi:https://doi.org/10.1016/j.psj.2020.06.043). However, in this study, the objective was different, as we wanted to compare the official ISO standard (which differentiates between C. jejuni and other Campylobacter species with the hippurate test) with the use of MinIon. The material and methods have been modified to clarify this point.
For Campylobacter Speciation, strains isolated were plated on Columbia Blood Agar (Oxoid Ltd., Basingstoke, UK) and incubated at 41.5 â—¦C for 44 ± 4 h in modified atmosphere (5% O2, 85% N2, 10% CO2, CampyGen, Oxoid, Basingstoke, UK). Finally, the hippurate hydrolysis test (Oxoid, Madrid, Spain) was used to determine the species of the Campylobacter (Lines 157-161)
Point 3: The information provided on the diversity and structure of caecal communities in birds at 14 and 42 days-of-age is of marginal value, and outside the scope of the study.
Response 3: The researchers of this study have been selected these two sampling dates because they are two key moments in the broiler production cycle. At 14d of age is when the broiler immune system starts to be mature. In addition, the 42 d of age is when the animals leave to the slaughterhouse.
To clarify this point a paragraph has been included in the Introduction (Lines 54-59): “In addition, it is important to choose the best moment to assess the Campylobacter status of the flock. There are two key moments in the broiler production cycle (14 and 42 days of age), since at 14d of age is when the broiler immune system starts to mature and the 42 d of age is when the animals leave for the slaughterhouse (Marin et al., 2009). For this reason, any decision taken at these two crucial moments can directly influence the broiler production cycle and the final product”.
Marin et al., 2009. Poultry Science 88 :1999–2005. doi: 10.3382/ps.2009-00040
Point 4: The number of samples examined is small (n=6) and limited to birds inoculated with C. jejuni. Expanding the study to actual production samples would be beneficial.
Response 4: The number of samples collected per pen is according to previous papers for this type of sample (Pineda-Quiroga et al., 2019; Montoro-Dasi et al., 2020). In addition, each sample consisted of a pool of 3 animals which reduces individual variability, estimates group-level microbiome diversity, and improves the efficiency of large-scale pathogen screening campaigns (Ray et al., 2019; Furstenau et al., 2020). However, we currently have a project with a more important budget, where based on the results obtained in this study, we are going to increase the number of animals and the pathogenic microorganisms studied (both for public health and animal health importance), as well as the control tools applied for these pathogens. For this purpose, we will always use both the official standard (ISO) and MinIon.
Furstenau TN, Cocking JH, Hepp CM, Fofanov VY. Sample pooling methods for efficient pathogen screening: Practical implications. PLoS One. 2020, 15(11):e0236849. doi: 10.1371/journal.pone.0236849.
Montoro-Dasi L, Villagra A, de Toro M, Pérez-Gracia MT, Vega S, Marin C. Fast and Slow-Growing Management Systems: Characterisation of Broiler Caecal Microbiota Development throughout the Growing Period. Animals (Basel). 2020, 10(8):1401. doi: 10.3390/ani10081401.
Pineda-Quiroga C, Borda-Molina D, Chaves-Moreno D, Ruiz R, Atxaerandio R, Camarinha-Silva A, García-Rodríguez A. Microbial and Functional Profile of the Ceca from Laying Hens Affected by Feeding Prebiotics, Probiotics, and Synbiotics. Microorganisms. 2019, 7(5):123. doi: 10.3390/microorganisms7050123.
Ray KJ, Cotter SY, Arzika AM, Kim J, Boubacar N, Zhou Z, Zhong L, Porco TC, Keenan JD, Lietman TM, Doan T. High-throughput sequencing of pooled samples to determine community-level microbiome diversity. Ann Epidemiol. 2019, 39:63-68. doi: 10.1016/j.annepidem.2019.09.002.
Point 5: Information on the relative cost (and merits) of the technology relative to other strategies (e.g. quantitative PCR) would be useful to include.
Response 5: According to the reviewer's suggestion, the information related to the cost of both techniques has been included in a new section within the material and methods (Lines 207-214) and results (311-316). In addition, a comparative table has also been included (Table 4). (Lines 317-320)
SPECIFIC COMMENTS
Point 6: L37. Specify the Campylobacter species (presumably C. jejuni) here and throughout.
Response 6: According to the suggestion of the reviewer, the specie jejuni has been included in the manuscript.
Point 7: L58. Anomalous hyphen. This was noted elsewhere as well.
Response 7: Hyphen has been removed
Point 8: L76. Were requisite biosafety and biosecurity standards met? This should be specified.
Response 8: The conditions under Royal Decree 53/2013 of 1 February must be met, which are guaranteed after approval by the Ethical Committee for Animal Experimentation. In this process, compliance with the 3Rs (Reduction, Replacement and Refinement) was ensured. In addition, in the development of the study, no procedures have been carried out that include the regulated techniques applied to the animal until it is slaughtered. The conditions in which the animals are housed are taken into account, with no modifications to commercial standards (the facility has a controlled environment that guarantees a temperature of 32ºC on day 1 of life until 19ºC on the last day of life (day 42), a humidity of 55±10% and 15-20 renewals per hour). The animals were handled according to good management practices to minimise animal stress and, prior to slaughter, stunning. The procedure does not lead to any decrease in activity or production performance as Campylobacter does not show any symptoms in the birds. Sampling of caecal contents was carried out under aseptic conditions, avoiding contamination of the samples.
Point 9: L88. Superscript m2.
Response 9: Superscripts have been corrected.
Point 10: L100. Indicating “spp.” in the subheading is misleading. I was expecting to see information on speciation of Campylobacter.
Response 10: According the reviewer suggestion the information is included in the manuscript.
Point 11: L102. It would be beneficial to include additional information on the strain of C. jejuni used. Moreover, determination that the strain recovered was the same as the inoculation strain should be determined (i.e. fingerprinted).
Response 11: The strain used in this study has been isolated from previous studies performed at the field level, and the reference has been included in the manuscript (Ingresa-Capaccioni et al., 2016).
To avoid the problem indicated by the reviewer, samples were collected both from the environment (before the arrival of the chicks) and from the chicks on arrival at the farm. Both environmental and flock status was negative until the animals' experimental infection.
Ingresa-Capaccioni S, Jiménez-Trigos E, Marco-Jiménez F, Catalá P, Vega S, Marin C. Campylobacter epidemiology from breeders to their progeny in Eastern Spain. Poult Sci. 2016, 95(3):676-83. doi: 10.3382/ps/pev338.
Point 12: L103. Additional information on how the inoculum was generated should be included.
Response 12: To generate the inoculum, an overnight inoculum was diluted to obtain a wavelength of 0.2 (OD 600, 108 CFU/mL). Then, ten-fold dilutions were made until a concentration of 106 CFU/mL, and animals were infected with 100 μL (105 CFU/bird).
Point 13: L117. Spraying the caeca with ethanol may kill C. jejuni, but it will not kill endospores. Thus, the sentence should be reworded.
Response 13: According with reviewer suggestion, the sentence has been re-written “Individual caecum samples were obtained and placed in sterile pots. Briefly, the surface of each intact caecum was spray treated with 70% ethanol as directed by Hansson et al. [38] before caeca contents were extracted.” (Line 128)
Point 14: L118. How the samples were homogenized should be specified. Were beads used? This is crucial to guard against bias due to differential homogenization of gram negative vs gram positive taxa.
Response 14: We fully agree with the reviewer that the beads technique is excellent for homogenising many kinds of samples. However, because this study is a proof of concept in the field, the equipment moved was minimal to mimic how it will be applied in reality. Moreover, it is the same homogenization technique used officially (ISO) as a control. The caecal contents of chickens are very fluid, so once all caecal contents were removed from the caecum and placed in a sterile Petri dish, they were homogenized vigorously with the help of a sterilized swab until a homogenized product was obtained.
This homogenization technique has been used previously in numerous laboratory studies, with optimal results (Montoro-Dasi et al., 2020:2021). The information has been included in the manuscript to avoid misunderstandings: “To this end, the caeca content was placed in a sterile Petri dish and homogenized vigorously with a sterile swab until a homogenized product was obtained (Montoro-Dasí, 2020:2021)”.
Montoro-Dasi L, Villagra A, de Toro M, Pérez-Gracia MT, Vega S, Marin C. Fast and Slow-Growing Management Systems: Characterisation of Broiler Caecal Microbiota Development throughout the Growing Period. Animals (Basel). 2020, 10(8):1401. doi: 10.3390/ani10081401.
Montoro-Dasi L, Villagra A, de Toro M, Pérez-Gracia MT, Vega S, Marin C. Assessment of Microbiota Modulation in Poultry to Combat Infectious Diseases. Animals (Basel). 2021, 11(3):615. doi: 10.3390/ani11030615.
Point 15: L136. Italicize Campylobacter here and throughout.
Response 15: Italics have been corrected
Point 16: L139. Going beyond oxidase and catalase tests is necessary. For example, utilization of taxon-specific PCR.
Response 16: The utilization of taxon-specific PCR is always interesting to characterize a strain, as we have published previously (Marin et al., 2020). However, as reported before, the researchers of this study used the EU official technique ISO 10272-2:2017, were the biochemical tests such as oxidase, catalase, etc. have been included.
Marín, C., Sevlla-Navarro, S., Lonjedo, R., Catalá-Gregori, P., Ferrús, M. A., Vega, S., Jiménez-Belenguer, A. Genotyping and Molecular Characterisation of Antimicrobial Resistance in Thermophilic Campylobacter Isolated from Poultry Breeders and Their Progeny in Eastern Spain; 2020;
Point 17: L179. “serotype prediction”?
Response 17: The term has been corrected (Lines 202: Campylobacter taxa prediction)
Point 18: L269. Similarly to L100, this subheading is misleading.
Response 18: With the sequencing method and analytical study employed we have not reached the specie level, so specifying jejuni would be an assumption.
Point 19: L343. “This workflow could help farmers …”. May be so, but the study scope could have better addressed this (see general comment above).
Response 19: The authors of this paper fully agree with the reviewer's comments and suggestions. Therefore, all of them will be taken into account for the Spanish-wide study, from which we hope to obtain promising data for the poultry production sector.
Reviewer 3 Report
I'm glad to see that a research group has conducted this line of work! The manuscript was well written and comprehensive. My only comments:
Line 202- I suspect "obtention" is not a commonly used word, perhaps rewrite that sentence as, "... extraction to a minimum limit of 239,988 reads..."
Line 203- I guess it's possible, but with such a narrow range for read length, I would not expect such a high standard deviation for the average.
Author Response
Dear Reviewer,
The researchers of this manuscript really appreciate your comments and suggestions. All the comments have been included in the manuscript.
Specific comments:
Point 1: Line 202- I suspect "obtention" is not a commonly used word, perhaps rewrite that sentence as, "... extraction to a minimum limit of 239,988 reads..."
Response: Sentence has been rewritten (Lines 233-234: The complete MinION sequencing process, starting from the DNA extraction to the ex-traction to a minimum limit of 239,988 reads, lasted 10.5 h)
Point 2: Line 203- I guess it's possible, but with such a narrow range for read length, I would not expect such a high standard deviation for the average.
Response: We have included the SD of the maximum and minimum length values to avoid making it look like an error.